# The Seasonality of Eddy-Induced Chlorophyll-a Anomalies in the Kuroshio Extension System

**Tongyu Wang [1,*], Shuwen Zhang [2], Fajin Chen [3] and Luxing Xiao [1]**

1   College of Geographical Sciences, Lingnan Normal University, 29 Cunjin Road, Zhanjiang 524048, China; xiaolx2023@163.com
2   Institute of Marine Science, Shantou University, Shantou 515063, China; zhangsw@stu.edu.cn
3   College of Ocean and Meteorology, Guangdong Ocean University, Haida Road, Zhanjiang 524091, China; fjchen@gdou.edu.cn
*   Correspondence: wangtongyu@lnsfxy.wecom.work

**Abstract:** The Kuroshio Extension (KE) System exhibits highly energetic mesoscale phenomena, but the impact of mesoscale eddies on marine ecosystems and biogeochemical cycling is not well understood. This study utilizes remote sensing and Argo floats to investigate how eddies modify surface and subsurface chlorophyll-a (Chl-a) concentrations. On average, cyclones (anticyclones) induce positive (negative) surface Chl-a anomalies, particularly in winter. This occurs because cyclones (anticyclones) lift (deepen) isopycnals and nitrate into (out of) the euphotic zone, stimulating (depressing) the growth of phytoplankton. Consequently, cyclones (anticyclones) result in greater (smaller) subsurface Chl-a maximum (SCM), depth-integrated Chl-a, and depth-integrated nitrate. The positive (negative) surface Chl-a anomalies induced by cyclones (anticyclones) are mainly located near (north of) the main axis of the KE. The second and third mode represent monopole Chl-a patterns within eddy centers corresponding to either positive or negative anomalies, depending on the sign of the principal component. Chl-a concentrations in cyclones (anticyclones) above the SCM layer are higher (lower) than the edge values, while those below are lower (higher), regardless of winter variations. The vertical distributions and displacements of Chl-a and SCM depth are associated with eddy pumping. In terms of frequency, negative (positive) Chl-a anomalies account for approximately 26% (18%) of the total cyclones (anticyclones) across all four seasons. The opposite phase suggests that nutrient supply resulting from stratification differences under convective mixing may contribute to negative (positive) Chl-a anomalies in cyclone (anticyclone) cores. Additionally, the opposite phase can also be attributed to eddy stirring, trapping high and low Chl-a, and/or eddy Ekman pumping. Based on OFES outputs, the seasonal variation of nitrate from winter to summer primarily depends on the effect of vertical mixing, indicating that convective mixing processes contribute to an increase (decrease) in nutrients during winter (summer) over the KE.

**Keywords:** mesoscale eddies; chlorophyll concentrations; biogeochemical cycling

## 1. Introduction

Mesoscale eddies, which encompass spatial scales ranging from 10 to 100 km and temporal scales from weeks to months, are significant drivers of complex interactions within the ocean, including its physics, biology, and biogeochemistry [1]. These eddies are commonly observed in west boundary currents and are characterized by strong vertical and horizontal currents. The Kuroshio Extension (KE), situated in the North Pacific Ocean, is a notable component of the western boundary current. It is distinguished by its elevated temperature, salinity, flow velocity, and noticeable ocean color [2].

The KE region is renowned for its prominent ocean front and the highest eddy kinetic energy globally. This unique oceanic area is influenced by various factors, including topography, monsoons, and the western boundary current. These factors contribute to the

manifestation of complex and dynamic characteristics observed in the eddies present within the KE [3]. Throughout the lifecycle of an eddy, from its formation to dissipation, it actively transports its internal water mass in a westward direction. This transport is facilitated by a range of physical processes, including eddy stirring, trapping, pumping, and eddy-wind Ekman pumping [4–10]. These processes play a crucial role in facilitating the horizontal exchange and vertical transport of physical and chemical constituents, such as heat, salt, kinetic energy, nutrients, and phytoplankton. The physical and biological processes within eddies has a significant impact on the ecological environment of the upper ocean [11–13]. The widely accepted eddy pumping process, based on satellite measurements of cyclonic (anticyclonic) eddies, suggests that the uplifting (deepening) of isopycnals and nitrate into (out of) the euphotic zone stimulates (inhibits) phytoplankton growth, resulting in elevated (reduced) concentrations of chlorophyll-a (Chl-a) [14–20]. Lin et al. [21] later successfully reproduced the above decadal variations with their coupled physical–biological model and conducted a mixed layer nitrogen budget analysis. It is shown that the vertical advection mainly caused by nutricline heaving plays an important role. However, their analyses mostly focused on the impacts of the decadal variations of the KE on the surface Chl-a in the annual mean and the spring bloom. Since Chl-a often retains its maximum in the subsurface layer in this region, it will be illuminating to also examine subsurface variability for more comprehensive understanding.

With the advancement of Argo and Bio-Argo float deployments, researchers now have effective tools to explore the subsurface dynamics within eddies without relying solely on satellite observations. Recent studies have highlighted the intensified surface Chl-a concentrations in anticyclonic eddies during winter, attributed to convective mixing stirring the subsurface Chl-a maximum [8]. Winter cyclonic eddies (CEs) have been found to promote enhanced vertical blooms dominated by phytoplankton biomass due to shallow mixed layers and uplifted thermoclines [9]. Similarly, intense downwelling within anticyclonic eddies (ACEs) in the South China Sea leads to a substantially deeper and less pronounced subsurface Chl-a maximum compared to CEs during summer [10]. Studies conducted in the North Subtropical Pacific have revealed depth-integrated nitrate and Chl-a anomalies within eddies, generally within ±90% for nitrate and ±10% for Chl-a in the euphotic layer, with magnitudes varying based on individual eddy analyses [12]. Rii et al. [22] and Bidigare et al. [23] reported substantial increases in anomalies within CEs compared to areas outside of eddies, with magnitudes ranging from 3.3 to 9.0 times and 1.0 to 1.5 times, respectively. These variations highlight the importance of incorporating a significant amount of subsurface observation data to distinguish the contributions of different physical processes. Furthermore, the spatial pattern of Chl-a concentrations can be strongly affected by different physical processes (eddy stirring, trapping, pumping, and eddy-wind Ekman pumping) under different seasons [1]. Due to the limited amount of data, traditional ship-based observations have been able to focus only on one eddy or on several specific eddies. These processes can have variable effects depending on the conditions within each eddy and the regional-scale circulation. Therefore, it is essential to quantify the effects of eddies on Chl-a concentrations across different regions and under various circumstances. Overall, previous studies have provided valuable observational evidence concerning regional variations in the biogeochemical response to eddies. Nevertheless, the biogeochemical response driven by eddies remains unclear, particularly with regard to different seasons.

To gain a comprehensive understanding of the biogeochemical responses driven by eddies in the KE, this study considers the seasonal variations in Chl-a concentrations and their spatial patterns. By integrating satellite data, oceanographic observations, and simulation outputs, this study aims to quantify the effects of eddies on Chl-a concentrations across different regions and under various circumstances. This comprehensive approach allows for a robust assessment of the impacts of eddy dynamics on the vertical distribution of Chl-a concentrations and facilitates a better understanding of the underlying mechanisms governing these variations.

This study examines the seasonal variations in eddy-driven biogeochemical responses in the KE using satellite data and decades of oceanographic observation data. Eddy-associated surface Chl-a anomalies spanning 21 years are composited onto eddy-centric coordinates. Additionally, Argo floats are used to normalize Chl-a, nutrient, temperature, and salinity measurements into a standardized vertical profile, allowing for a statistical analysis of the vertical structure (physical and biological) of eddies in the KE under different seasons. OFES outputs are used to evaluated the contribution of mesoscale dynamics to nutrient transport.

## 2. Materials and Methods

### 2.1. Materials

#### 2.1.1. In Situ Data

In situ data are obtained from global Argo project (Argo and Bio-Argo) (downloaded from http://www.argo.net (accessed on 15 January 2022)), which are included the profile of temperature, salinity, Chl-a, nutrient, and dissolved oxygen. The data are quality-controlled using automated procedures and assessed using statistical analysis residuals. The international Argo program has recognized the importance of ensuring the quality of Argo float observation data from its early stages and has implemented a series of measures to achieve quality control and data standardization. Here, we extracted the in situ data from 1998 to 2020 in the KE to reveal the change of vertical physical and biochemical structures, approximately 19,000 temperature and salinity profiles, 294 nitrate profiles, and 261 Chl-a profiles (Table 1). Argo floats (Chl-a, nutrient, temperature, and salinity) are normalized into a standard vertical profile under different seasons, which provided a statistical view of the vertical structure (biological and physical) of eddies in the KE.

**Table 1.** Numbers of Argo floats under different seasons.

| | CE | | | | | ACE | | | | | Edge | | | | |
|---|---|---|---|---|---|---|---|---|---|---|---|---|---|---|---|
| | Winter | Spring | Summer | Autumn | All | Winter | Spring | Summer | Autumn | All | Winter | Spring | Summer | Autumn | All |
| Chl-a | 11 | 11 | 9 | 3 | 34 | 29 | 10 | 65 | 88 | 192 | 10 | 8 | 4 | 13 | 35 |
| Nitrate | 15 | 32 | 16 | 4 | 67 | 36 | 24 | 44 | 43 | 147 | 29 | 28 | 7 | 16 | 80 |
| Temperature and Salinity | 907 | 1377 | 922 | 474 | 3680 | 1447 | 1069 | 2008 | 2623 | 7147 | 2364 | 2808 | 1818 | 2056 | 9046 |

#### 2.1.2. Satellite Observations

The daily merged satellite product of Chl-a concentration data was obtained using a multisensor approach that integrated data from various sensors, such as SeaWiFS, MODIS, MERIS, VIIRS-SNPP, JPSS1, OLCI-S3A, and S3B. These algorithms were designed to provide accurate estimations with a spatial resolution of 4 km, catering to the needs of end-users. The Chl-a concentration data from the merged satellite product underwent rigorous processing and validation procedures to ensure their accuracy and reliability. These procedures involved calibrating and cross-comparing data from multiple sensors, applying atmospheric corrections, and validating the results against in situ measurements and independent validation datasets. The Copernicus Marine Environmental Monitoring Center (CMEMS, http://marine.copernicus.eu/ (accessed on 12 January 2022)) follows strict quality control measures to ensure that the data meet high standards. The CMEMS served as the source for acquiring the data, which was utilized to analyze and characterize the biomass dynamics at the sea surface [24]. For the analysis of eddies, sea level anomaly (SLA) data were obtained from the CMEMS. These data had a spatial resolution of 25 km and a temporal resolution of 1 day. Multimission altimetry datasets were used to detect the eddies. Each day, the detected eddies provided information on their location, type (cyclonic or anticyclonic), speed, radius, and associated metadata. The detection process employed a delayed-time algorithm (version DT2.0exp), which was developed and validated in collaboration with D. Chelton and M. Schlax at Oregon State University [1]. The altimetry period covered data from 1993 to the present. By utilizing these datasets and detection methods,

this study aims to investigate the characteristics and behavior of eddies in the designated region.

### 2.1.3. Model Data

The nutrient levels in the upper ocean, specifically above 200 m, were characterized by utilizing multiyear average profiles of nitrate. These profiles were derived from the World Ocean Atlas 2018 (WOA18), which can be accessed at https://www.nodc.noaa.gov/ocs/woa18 (accessed on 20 January 2021).

The OFES quasi-global eddy-resolving simulation (accessible at http://www.jamstec.go.jp/ofes/ofes.html (accessed on 20 January 2021)) utilized the Modular Ocean Model ver.3 (MOM3), developed by GFDL/NOAA, with a horizontal resolution of 0.1 degree and 54 vertical levels featuring varying intervals from 5 m at the surface to 330 m at a maximum depth of 6065 m. By combining the model with a 5-year integration of NCEP/NCAR monthly climatology, it successfully captured the annual cycle of temperature, nitrogen (the sole nutrient), and phytoplankton in the upper ocean [20]. To ensure accurate representation of oceanic conditions, the model is combined with a 5-year integration of NCEP/NCAR monthly climatology. This integration successfully captures the annual cycle of temperature, nitrogen (the sole nutrient), and phytoplankton in the upper ocean. The model outputs, including monthly velocity and nitrate fields, provide valuable information for investigating the influence of mesoscale dynamics on biological production in the study region. The data from the OFES simulation offer a monthly time resolution, allowing for the examination of the seasonal variations and long-term trends in mesoscale dynamics and their impact on biological processes. The simulation results are of high quality, as the model has been extensively validated against observational data and has demonstrated skill in reproducing key features of the oceanic circulation and its biogeochemical properties.

### 2.2. Methods

### 2.2.1. Composite Analysis

In this study, a composite analysis was performed using the normalized Chl-a anomalies (*Chla'*) field. The Chla' field was obtained by collocating and projecting the corresponding normalized Chl-a anomalies onto a high-resolution grid with normalized eddy-centric coordinates. The grid had a resolution of 0.025 and consisted of a 161 × 161 grid of cells. The size of the collocation region was determined based on the eddy radius, denoted as R, to ensure accurate spatial alignment and analysis. The *Chla'* values were calculated for each individual eddy in the composite analysis. By utilizing the composite analysis approach, the study obtained the mean Chla' for each eddy through grid summation. This enabled an effective analysis and quantification of the *Chl-a'* associated with eddies in the study area. *Chla'* were used for each composite individual eddy and calculated as:

$$Chla' = \frac{Chl-a - \overline{Chl-a}}{\sigma(Chl-a - \overline{Chl-a})} \tag{1}$$

where $\overline{Chl-a} = \frac{1}{4\pi R^2} \int_0^{2\pi} \int_0^{2R} Chl-a \cdot r dr d\theta$, the overbar denotes the radial average and $\sigma$ is the standard deviation, both computed over the eddy for r $\leq$ 2R.

### 2.2.2. EOF Analysis

Empirical orthogonal function (EOF) analysis was employed to identify dominant coherent variations and *Chl-a'* patterns within eddies across different seasons [25]. The space–time *Chl-a* data set A (*x, y, t*) can be expressed as:

$$A(x, y, t) = \sum_k a_k(t) F_k(x, y) \tag{2}$$

where $a_k(t)$ is the modulated function to show the corresponding temporal variation and $F_k(x, y)$ is the Kth model spatial patterns. Typically, the first three modes, characterized

by higher variance, are likely to possess significant physical interpretations [26]. Firstly, the *Chl-a'* data for each individual eddy in different seasons were collected. Then, these datasets were standardized to remove the effects of different scales and magnitudes. Next, the covariance matrix of the standardized *Chl-a'* datasets was computed. The EOFs were obtained by solving the eigenvalue problem of the covariance matrix. The eigenvalues represent the amount of variance explained by each EOF mode, while the eigenvectors represent the spatial patterns associated with each mode.

### 2.2.3. Mixed Layer Depth

Mixed layer depths (MLD(m)) were calculated using temperature and salinity data from Argo floats profiles. MLD were based on two criteria: the temperature change is 0.2 °C relative to the value at 10 m depth, and the density change is 0.03 kg/m$^3$ relative to the value at 10 m depth [27]. For each profile, the mean of the two values was chosen for MLD.

### 2.2.4. Mixed Layer Nutrient Budget

To explore the influence of mesoscale dynamics on biological production, we examined the nitrate budget equation based on the formulation introduced by Caniaux and Planton [28] for the mixed layer heat budget. The rate of change of nitrate ($N$) within the mixed layer [29] is:

$$\frac{\partial N}{\partial t} = -u_H \cdot \nabla_H N - u_Z \cdot \nabla_Z N + \frac{\partial K_z \partial N}{\partial_z{}^2} \tag{3}$$

The formula consists of three terms on the right side, namely lateral advection, vertical advection, and vertical mixing. The supply of nitrate, $N$, to the mixed layer is thus a suitable indicator for use in Equation (3) and can be inferred through lateral advection, vertical advection, and vertical mixing. Following the Reynolds averaging method, the lateral and vertical advective supply of $N$ can be separated into contributions from the mean flow and from the fluctuating eddy flow [30]:

$$\int\limits_{80}^{0} \overline{(-u_H \cdot \nabla_H N)}dz = \int\limits_{80}^{0} \left( \overline{-u_H} \cdot \overline{\nabla_H N} \right)dz + \int\limits_{80}^{0} \overline{(-u'_H \cdot \nabla_H N')}dz \tag{4}$$

$$\int\limits_{80}^{0} \overline{(-w \cdot \nabla_Z N)}dz = \int\limits_{80}^{0} \left( \overline{-w} \cdot \overline{\nabla_Z N} \right)dz + \int\limits_{80}^{0} \overline{(-w' \cdot \nabla_Z N')}dz \tag{5}$$

where $N$ is the concentration of the nitrate and w is the vertical velocity. The overbar denotes a time mean to be defined and the prime all deviations from this time mean (referred as the eddy term), and the subscripts $H$ and $Z$ represent "lateral" and "vertical", respectively. In this work, both the physical and biological variables (e.g., velocity fields and nitrogen) were adopted from OFES model outputs.

## 3. Results

This study focused on the KE region, specifically the area between 30°N and 40°N and 140°E and 170°E, as illustrated in Figure 1. The standard deviation (STD) of the average SLA within this region, calculated from data spanning from 1998 to 2020, exhibited high variability along the main axis of the KE at 35°N. Both the north and south sides of this axis displayed a substantial STD value of 0.3 m, indicative of pronounced eddy activity. The region adjacent to the main axis predominantly featured ACEs on the north side and CEs on the south side. Notably, the concentration of Chl-a within CEs was significantly higher compared to ACEs [11].

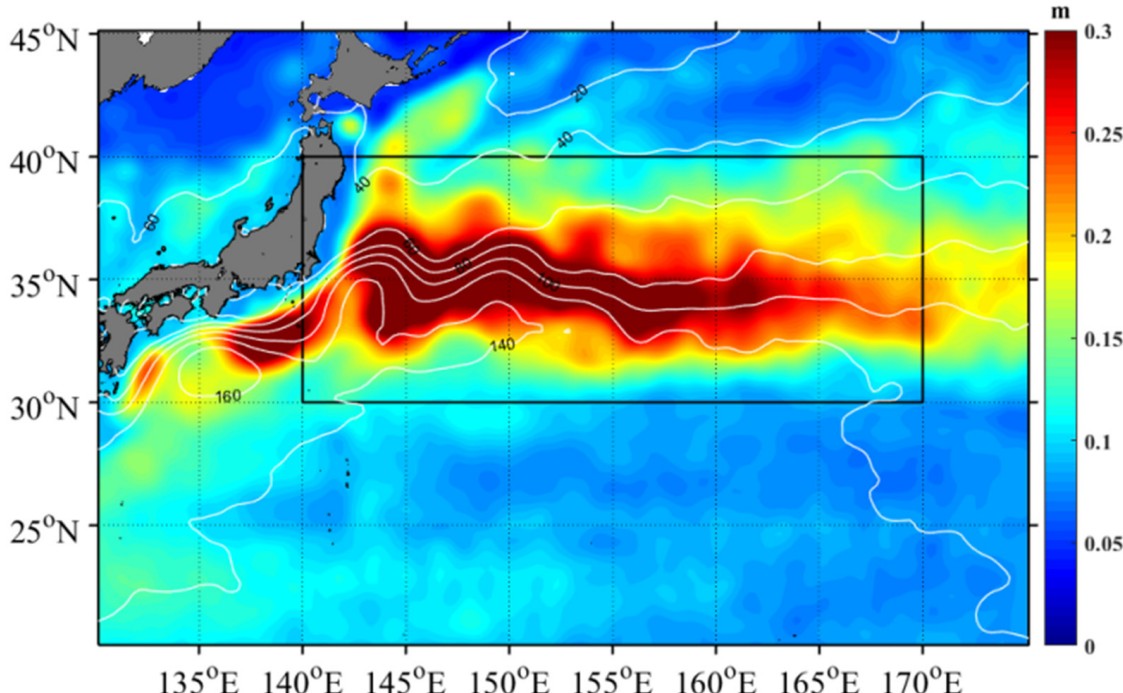

**Figure 1.** Spatial distribution of standard deviation for SLA from 1993 to 2020.

### 3.1. Eddy Properties

Table 2 presents the eddy properties in the KE during different seasons from 1998 to 2020. The table includes information on the number, amplitude, and rotational speed of CEs and ACEs. The activity of both CEs and ACEs shows similar seasonal variations, with the highest activity observed in spring. While ACEs generally have a larger radius across all seasons, CEs exhibit slightly higher values in terms of number, average amplitude, and rotational speed compared to ACEs. CEs are characterized by their smaller size and greater intensity compared to ACEs. Specifically, in winter, the number, amplitude, and rotational speed of CEs are 3%, 7%, and 11% larger than those of ACEs, respectively, while the radius of ACEs is 13% smaller than that of CEs. Additionally, we utilize a dimensionless coefficient (U/C, where U represents rotational speed and C denotes translational speed) greater than 1 to identify nonlinear eddies. This criterion indicates that the water within a certain depth maintains its distinct properties without exchanging with the surrounding environment [31]. Both CEs and ACEs exhibit a high level of nonlinearity according to this criterion [1].

**Table 2.** Statistics of eddy properties in different seasons of the KE.

| | CE | | | | | ACE | | | | |
|---|---|---|---|---|---|---|---|---|---|---|
| | Winter | Spring | Summer | Autumn | All | Winter | Spring | Summer | Autumn | All |
| Number | 574 | 630 | 594 | 547 | 1223 | 556 | 619 | 555 | 526 | 1180 |
| Amplitude (cm) | 14.04 | 13.71 | 14.55 | 15.2 | 14.38 | 13.12 | 13.1 | 13.77 | 13.66 | 13.41 |
| Radius (km) | 79.93 | 78.27 | 78.42 | 80.74 | 79.34 | 84.42 | 82.67 | 85.21 | 86.36 | 84.67 |
| Rotational Speed (m/s) | 0.36 | 0.37 | 0.39 | 0.39 | 0.38 | 0.32 | 0.33 | 0.33 | 0.33 | 0.33 |
| Chl-a anomaly (mg/m$^3$) | 0.48 | 0.41 | 0.31 | 0.11 | 0.33 | −0.5 | −0.37 | −0.36 | −0.29 | −0.38 |

### 3.2. Relationship between Eddy and Surface Chl-a

The relationship between SLA and Chl-a concentrations in the KE region was investigated to gain insights into the impact of eddies on surface Chl-a levels. Figure 2 illustrates

a notable negative correlation (r < 0.8, *p* < 0.05) between SLA and Chl-a, predominantly observed along the KE region (from 140°E to 180°E along 35°E) throughout the four seasons. The areas exhibiting a significant correlation align with the regions of high SLA STD, as depicted in Figure 1. These correlation patterns are in line with findings from previous studies [5,11]. The negative correlations can be attributed to the upwelling (downwelling) processes within CEs (ACEs), which transport nutrient-rich (-poor) water from deeper (shallower) regions, respectively.

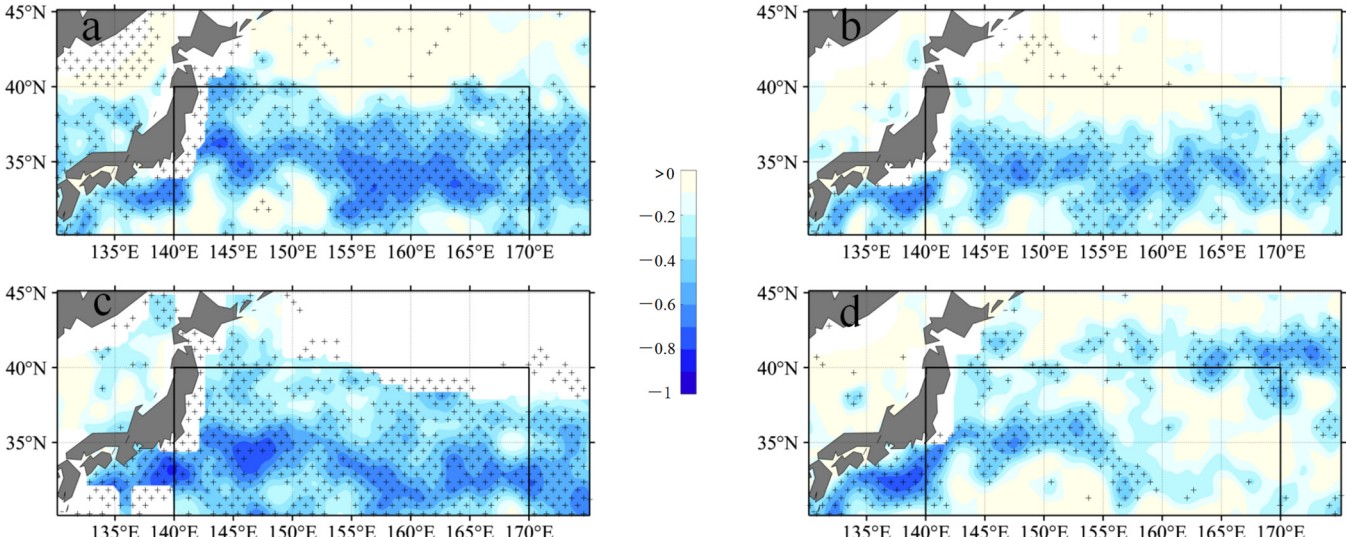

**Figure 2.** Spatial distribution of correlation (**a**) winter; (**b**) spring; (**c**) summer; and (**d**) autumn between SLA and Chl-a anomaly corresponding to eddies from 1993 to 2017. Plus sign denotes significant correlations at a 95% confidence level.

The spatial distribution of composite average normalized Chl-a concentrations across multiple eddies during different seasons provides detailed insights into the Chl-a response to CEs and ACEs (Figure 3). Monopole Chl-a patterns are observed only in winter and spring, indicating that higher (lower) Chl-a values within CEs (ACEs) during all four seasons correspond to positive (negative) Chl-a anomalies in the eddy center (r/R < 1) (Table 2). According to Table 2, the average Chl-a anomaly concentration in CEs (ACEs) is 0.48 (−0.5), 0.41 (−0.37), 0.31 (−0.36), and 0.11 (−0.29) mg/m$^3$ during different seasons, respectively. These positive (negative) anomalies in CEs (ACEs) can be attributed to eddy pumping, which is associated with upwelling (downwelling) processes.

Additionally, the large-scale horizontal gradient of Chl-a plays a significant role in the spatial patterns observed across all four seasons. The meridional advection of Chl-a contributes to the generation of Chl-a anomalies along the edges of eddies [8], particularly during summer and autumn. Moreover, the clockwise (anticlockwise) rotation of eddy flows leads to the accumulation of Chl-a anomalies in the northwest (southeast) quadrant of the eddies, influenced by the water transport in eddy-centric coordinates. The EOF method was employed to decompose normalized Chl-a data across thousands of eddies into significant temporal and spatial components that capture the primary variability in the spatial patterns. Figures 4 and 5 present the variances of the first three temporal modes for different seasons, illustrating the significant spatial variability captured by the first three spatial EOF modes. These figures also display the corresponding spatial EOF modes, which account for approximately 50% of the total variance in Chl-a for both CEs and ACEs. The first modes (variance: 18%~31%) correspond to the horizontal eddy stirring process, exhibiting Chl-a anomalies at the eddy edges. The second mode (variance: 12%~16%) represents monopole Chl-a patterns within the eddy centers, displaying either positive or

negative anomalies based on the sign of the principal component. Moreover, most of the third mode (variance: 10%~14%) exhibits similarities to the second mode.

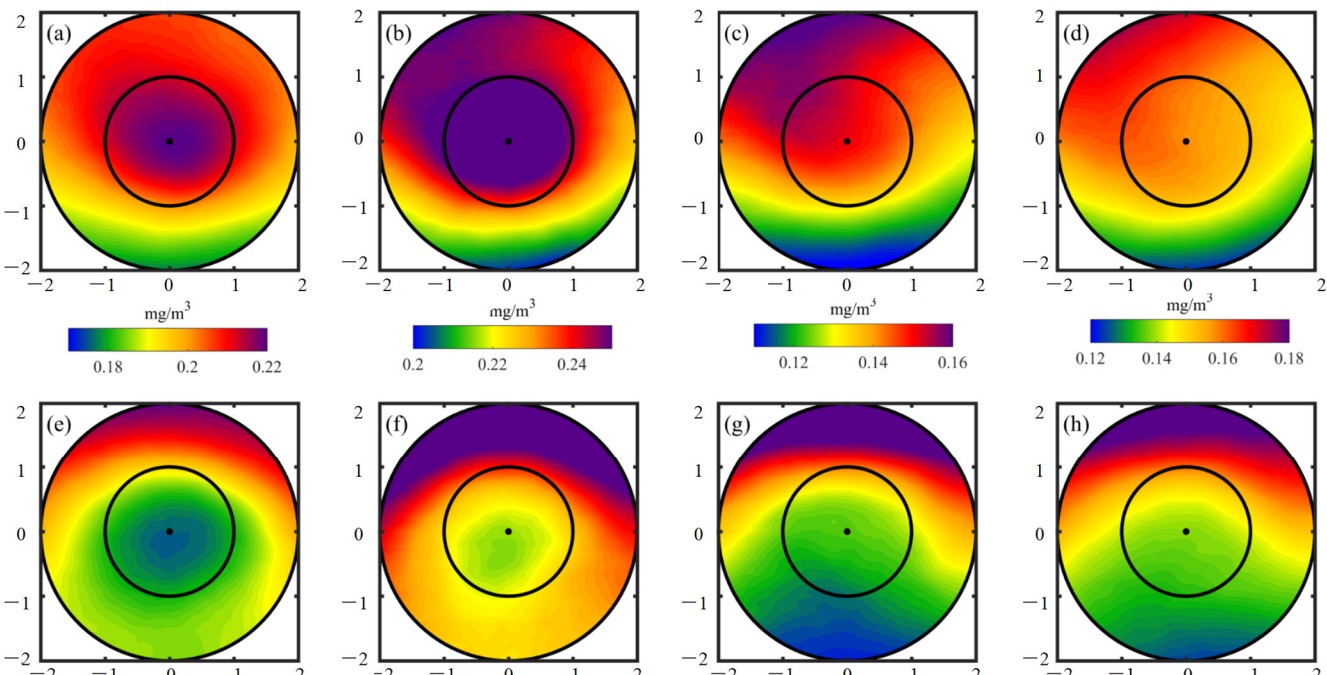

**Figure 3.** Composite averages of winter (**a,e**), spring (**b,f**), summer (**c,g**), and autumn (**d,h**). Chl-a associated with cyclonic eddies and anticyclonic eddies in the KE during the period 1998 to 2016. Inner and outer circles, respectively coincide with r/R < 1 and r/R < 2.

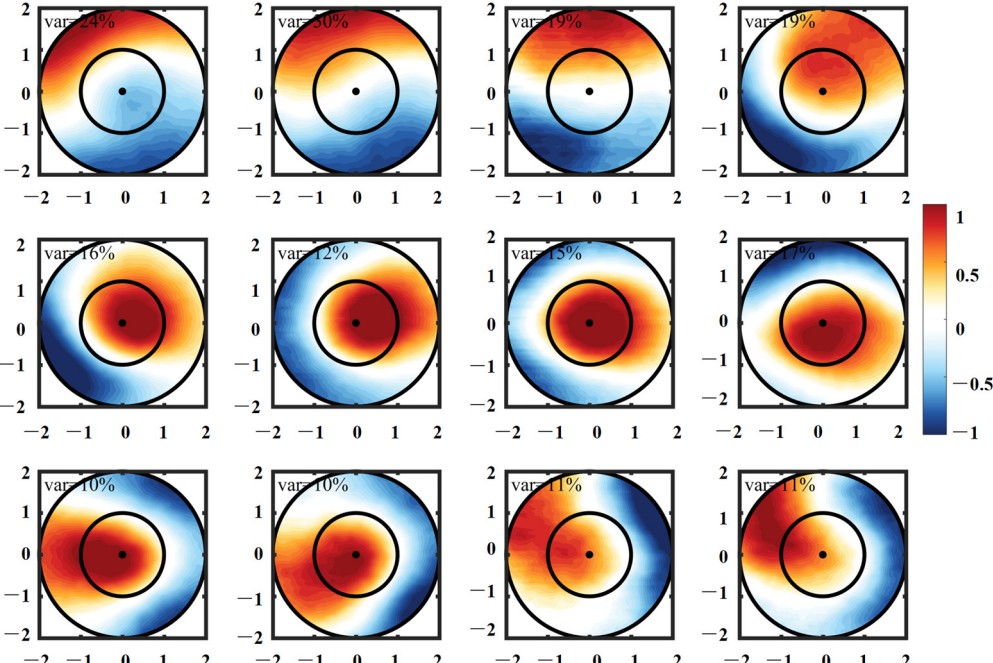

**Figure 4.** The first three spatial empirical orthogonal function (EOF) modes of Chl-a within cyclonic eddies in winter (**first column**), spring (**second column**), summer (**third column**), and autumn (**fourth column**).

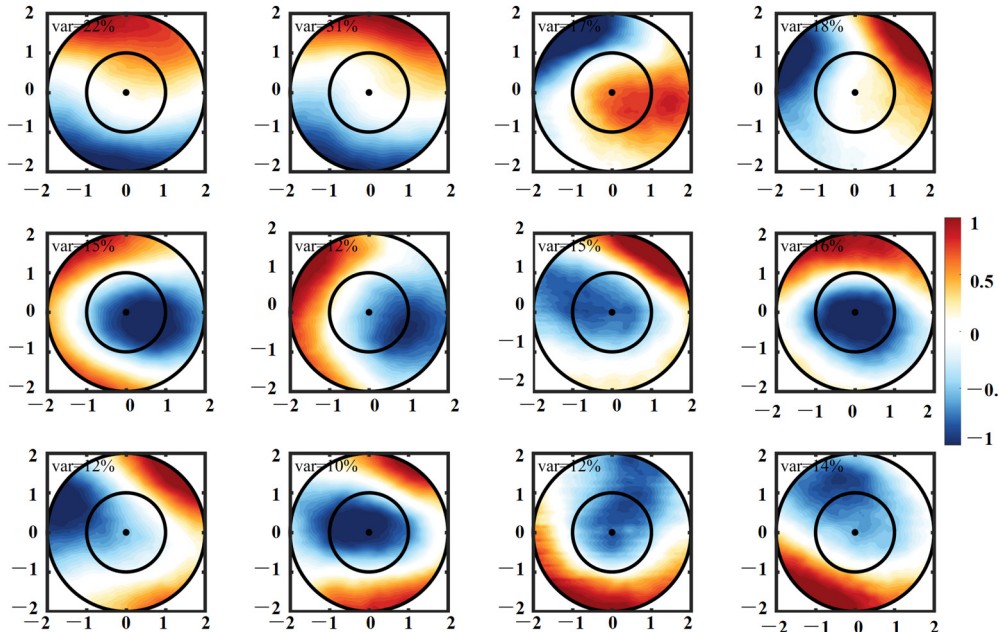

**Figure 5.** The first three spatial empirical orthogonal function (EOF) modes of Chl-a within anticyclonic eddies in winter (**first column**), spring (**second column**), summer (**third column**), and autumn (**fourth column**).

Figure 3 illustrates the composite averages of Chl-a anomalies associated with CEs and ACEs in the KE region. CEs exhibit positive Chl-a anomalies, while ACEs display negative Chl-a anomalies. However, it is important to note that not all eddies in the KE region exhibit Chl-a anomalies, as revealed by the EOF modes. Figures 4 and 5 depict the variability in Chl-a values within CEs and ACEs, respectively. Similar patterns have been observed in subtropical gyres, where ACEs have been found to enhance Chl-a concentrations through the modulation of winter mixing by eddies, as suggested by Dufois et al. [8].

To calculate the Chl-a anomalies associated with CEs and ACEs, each eddy is individually assessed and mapped onto a 0.5° × 0.5° grid. The results demonstrate that CEs generally increase Chl-a concentrations in most regions of the KE, while ACEs tend to decrease them. However, the magnitude of Chl-a anomalies varies across different regions. The spatial distribution of eddies also exhibits varying polarities. CEs induce higher positive Chl-a anomalies, primarily concentrated near the main axis of the KE. Conversely, negative anomalies are predominantly observed to the south of the main axis, while lower positive anomalies occur to the north. These negative anomalies account for 24% (winter), 23% (spring), 25% (summer), and 44% (autumn) of all CEs (Figure 6a,d,g,j). On the other hand, ACEs induce higher negative anomalies primarily north of the main axis, while positive anomalies are predominantly observed to the south. The positive anomalies account for 19% (winter), 21% (spring), 16% (summer), and 20% (autumn) of all ACEs (Figure 6b,e,h,k).

Previous research conducted by Itoh and Yasuda [32] has indicated that CEs are more commonly observed in the southern region of the KE compared to the northern region. This spatial distribution of eddies suggests that eddy activity may play a role in the regional variations observed in Chl-a anomalies associated with CEs and ACEs. Specifically, CEs located near the KE are more likely to encounter water with higher concentrations of Chl-a and nutrients originating from the northern region of the KE.

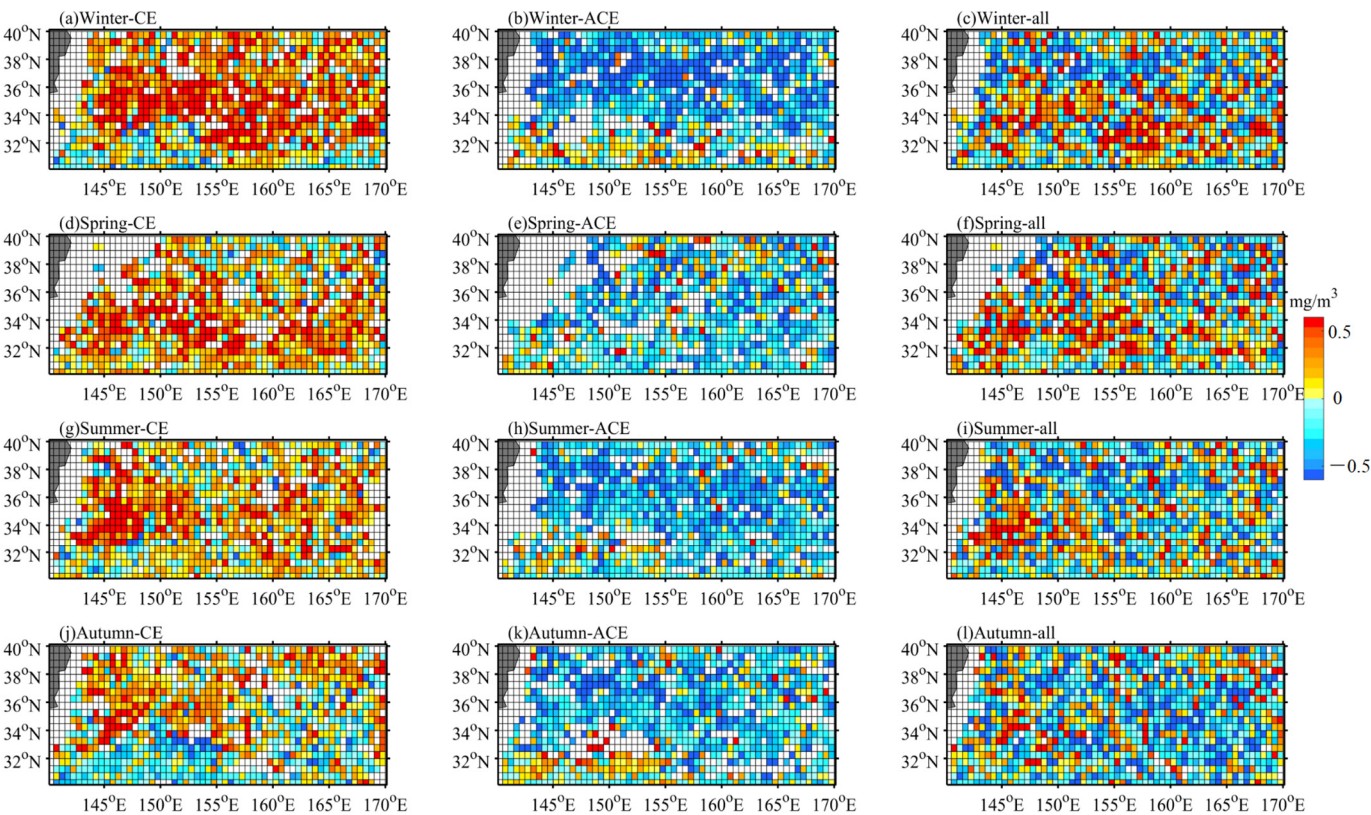

**Figure 6.** Mean regional winter (**a**–**c**), spring (**d**–**f**), summer (**g**–**i**), and autumn (**j**–**l**) Chl-a anomalies within cyclonic eddies and anticyclonic eddies in the KE during the period 1998 to 2016. Quantified at each 0.5 × 0.5 pixel.

### 3.3. Vertical Profiles of Temperature, Chl-a, and Nitrate in Eddies

An analysis was conducted on eddy trajectories and the distribution of temperature, salinity, nitrate, and Chl-a profiles obtained from the Argo program in the KE region. A total of approximately 19,000 temperature and salinity profiles, 294 nitrate profiles, and 261 Chl-a profiles were analyzed (refer to Table 1). Among these, 3680 (7100) temperature/salinity profiles were associated with CEs (ACEs), while 34 (192) Chl-a profiles and 64 (147) nitrate profiles were observed in CEs (ACEs). Additionally, 9000 temperature/salinity profiles, 35 Chl-a profiles, and 140 nitrate profiles were identified at the edges of the eddies (the edge is defined as the area of between 2R and 4R). To facilitate analysis, the Chl-a, nutrient, temperature, and salinity data were normalized into standardized vertical profiles corresponding to different seasons (Figures 7–9).

The results showed distinct characteristics of temperature, Chl-a distribution, and nutrient concentrations within eddies. Temperature within CEs was lower, while ACEs exhibited higher temperatures compared to the eddy edges across all seasons (Figure 7). During the colder seasons (winter and spring), the average temperature ranges between 10 and 20 °C. However, in the warmer seasons (summer and autumn), there is a noticeable increase of approximately 10 °C in the temperature within the 0–100 m depth range. The MLD experiences different patterns in CEs and ACEs. The MLD deepens to 90 m (120 m) due to winter deep mixing in CEs (ACEs), while the MLD shoals to 10 m (20 m) in CEs (ACEs) owing to seasonal stratification in summer. The variations in MLD within the same season indicate the presence of upwelling and downwelling processes, which contribute to vertical fluctuations within eddies and influence the vertical flux of nutrients and Chl-a.

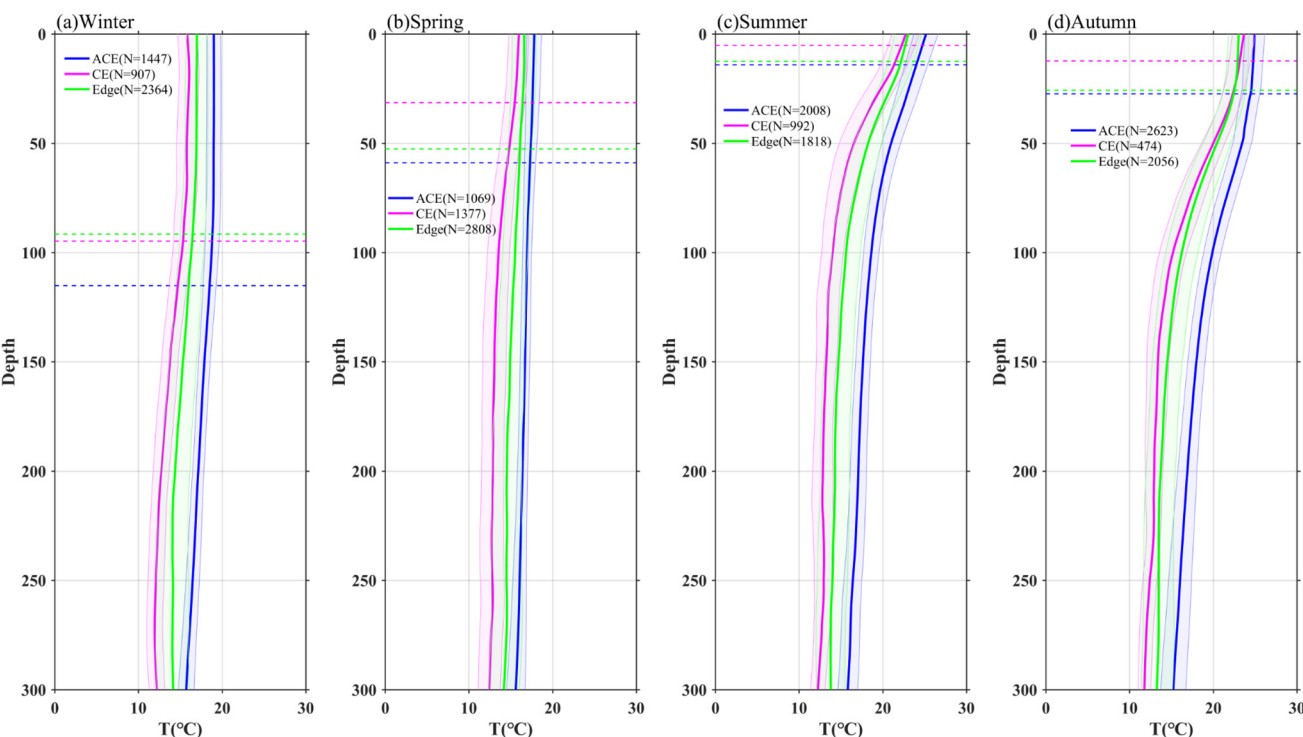

**Figure 7.** Vertical profiles of temperature in eddies during winter (**a**), spring (**b**), summer (**c**), and autumn (**d**). Composite mean vertical profiles for cyclonic eddies (pink), anticyclonic eddies (blue), and edges (green). N indicates the number of profiles. The shading represents standard errors. Dotted line represents mixed layer depth.

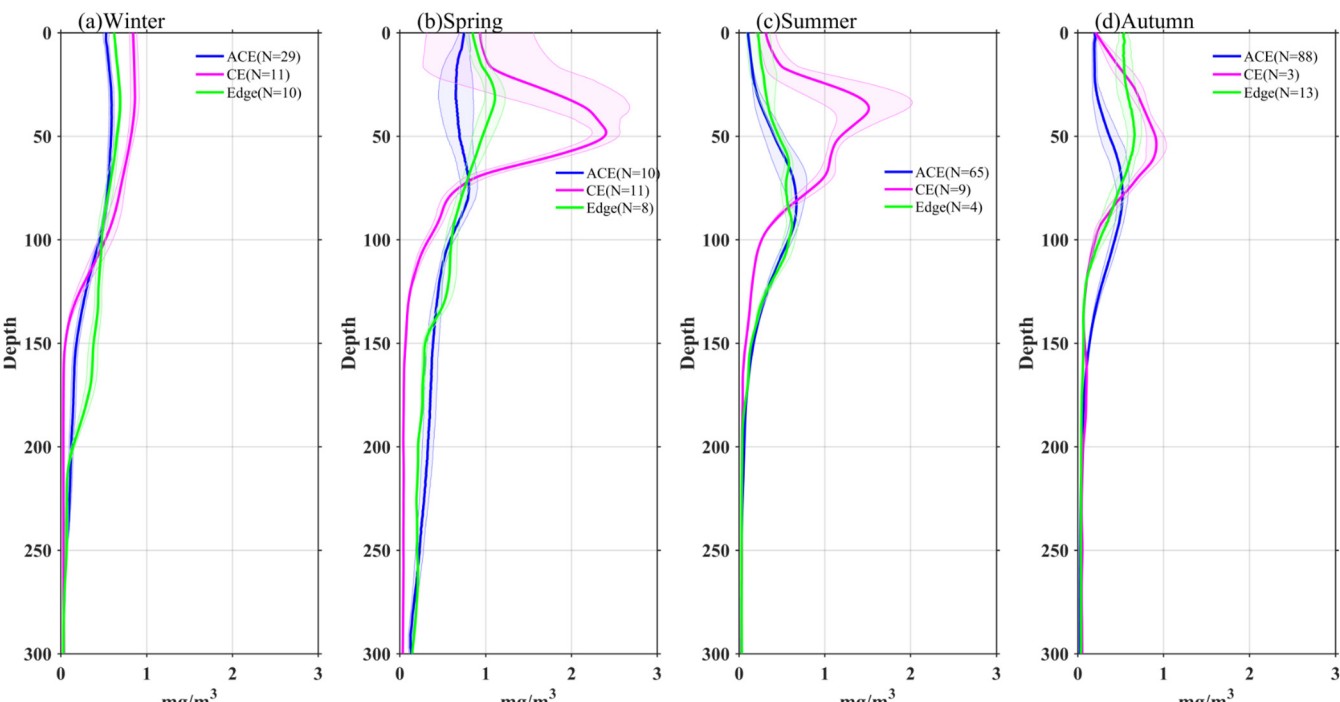

**Figure 8.** Vertical profiles of Chl-a in eddies during winter (**a**), spring (**b**), summer (**c**), and autumn (**d**). The shading represents standard errors.

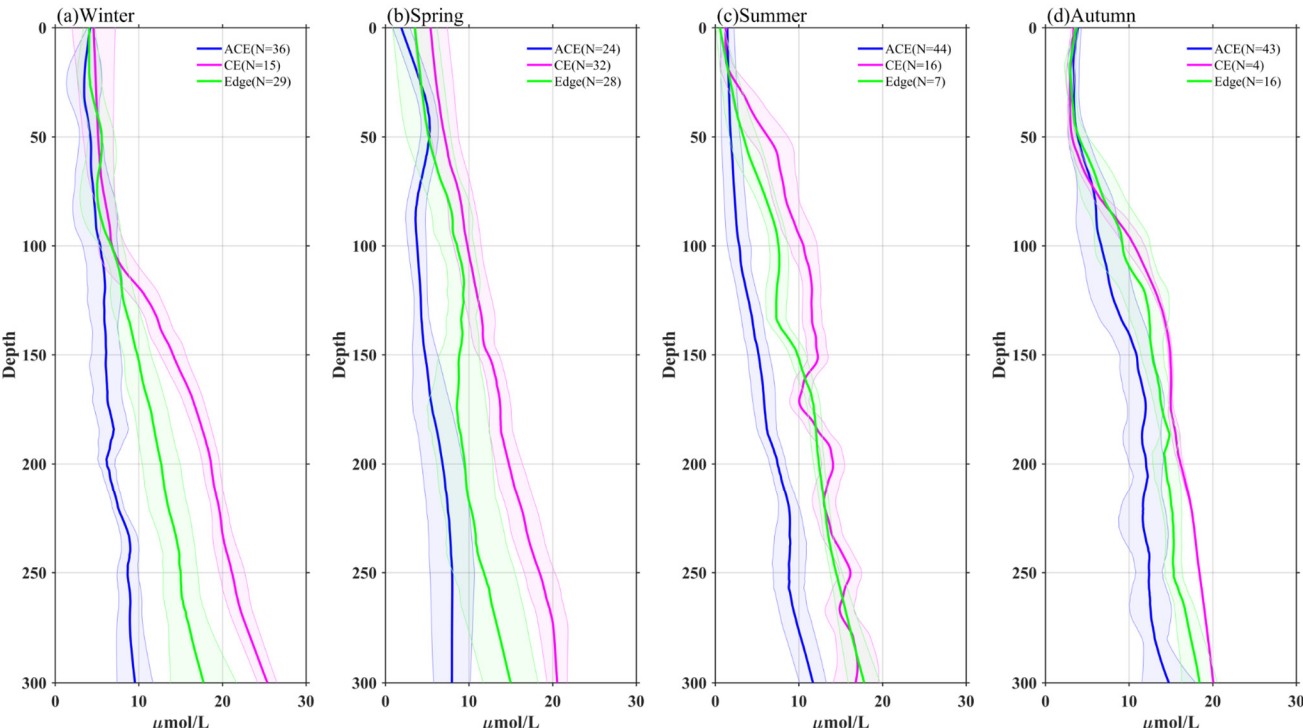

**Figure 9.** Vertical profiles of nitrate in eddies during winter (**a**), spring (**b**), summer (**c**), and autumn (**d**). The shading represents standard errors.

Regarding the vertical distribution of Chl-a within eddies (Figure 7), near-surface concentrations were higher in CEs compared to the eddy edges, while ACEs exhibited lower near-surface Chl-a concentrations. This pattern is consistent with the variations observed in near-surface Chl-a from satellite data (Figure 3), except for autumn. Additionally, the depth of the subsurface Chl-a maximum (SCM) layer is shallower in CEs than in ACEs. Specifically, the SCM depths in CEs are 48 m (spring), 36 m (summer), 54 m (autumn), and 30 m (winter), while in ACEs they are 75 m (spring), 81 m (summer), 78 m (autumn), and 35 m (winter). It is important to note that the vertical distribution of Chl-a above 100 m is relatively uniform due to winter mixing. The SCM within CEs reaches values of 0.87 mg/m$^3$ (winter), 2.3 mg/m$^3$ (spring), 1.5 mg/m$^3$ (summer), and 0.91 mg/m$^3$ (autumn), which are significantly higher compared to the edge values. In contrast, the SCM values in ACEs are 0.59 mg/m$^3$ (winter), 0.82 mg/m$^3$ (spring), 0.7 mg/m$^3$ (summer), and 0.52 mg/m$^3$ (autumn). Above the SCM layer, Chl-a concentrations are higher within CEs and lower within ACEs compared to the values at the eddy edges (Figure 8a–d). Conversely, below the SCM layer, Chl-a concentrations are lower within CEs and higher within ACEs than the edge values (Figure 8b–d), regardless of seasonal variations. These vertical distributions and displacements of Chl-a and SCM depth are associated with upwelling in CEs and downwelling in ACEs. These observations are consistent with the enrichment of nutrients within CEs and the depletion of nutrients within ACEs (Figure 9).

Furthermore, by comparing the profiles of chlorophyll-a and nitrate, it is observed that, as depth increases, the Chl-a concentration reaches its maximum within the subsurface layer (euphotic zone), while the corresponding nitrate concentration consistently decreases. This pattern is attributed to the rapid biological removal processes occurring within the euphotic layer [12,33,34], which make it challenging to detect significant variations in nitrate concentration associated with high Chl-a within the SCM layer. The depth-integrated nitrate over the upper 150 m was found to be higher within CEs and lower within ACEs compared to the edge values in each season (Figure 9). Specifically, it was 13% (winter), 25% (spring), 50% (summer), and 7% (autumn) higher within CEs, and 23% (winter), 37% (spring), 50% (summer), and 20% (autumn) lower within ACEs. The depth-integrated Chl-a

also exhibited a similar pattern, being higher within CEs and lower within ACEs compared to the edge values. It was 25% (winter), 42% (spring), 90% (summer), and 13% (autumn) higher within CEs, and 9% (winter), 19% (spring), 5% (summer), and 24% (autumn) lower within ACEs. These variations can be attributed to the vertical movement of isopycnals and nitrate, influencing phytoplankton growth and thus affecting the depth-integrated Chl-a within CEs and ACEs.

Regarding the seasonal variations, it was found that, in winter, despite high nutrient levels, reduced light availability limits a substantial increase in depth-integrated Chl-a within eddies. The changes in the vertical distribution of Chl-a during winter result from the redistribution of pre-existing Chl-a through enhanced vertical mixing. The higher depth-integrated Chl-a during the spring bloom can be explained through various hypotheses, including the critical depth, shoaling mixing layer, critical turbulence, onset of stratification, and disturbance-recovery hypotheses [35]. In summer, the depth-integrated Chl-a decreases but remains higher than in autumn, which can be attributed to high light availability and the influence of mesoscale modulation of the mixed layer depth on the persistence of high Chl-a/nitrate concentrations originating from the spring period, resulting in a more prominent SCM than in autumn.

## 4. Discussion

In this study, we observed that CEs in the KE region exhibit higher surface Chl-a concentrations compared to the edge values, while ACEs show lower surface Chl-a concentrations. This pattern is particularly pronounced during the cold seasons (winter and spring) when Chl-a levels are typically at their highest. The analysis of Chl-a patterns within eddies, using EOF analysis, has revealed the complexity of these relationships, indicating that multiple mechanisms contribute to the interaction between eddies and Chl-a dynamics. Nutrients and light availability are crucial factors influencing phytoplankton blooms. Eddies play a significant role in shaping the vertical distribution of Chl-a and nitrate through processes such as eddy pumping, eddy Ekman pumping, eddy stirring, variations in light conditions, and vertical mixing, which vary across different seasons [12,16,36].

### 4.1. Modulation of Mixed Layer Depth by Eddies

The modulation of the MLD by eddies has also been identified as a contributing factor to elevated Chl-a levels [8,18]. In our study, we followed the approaches of previous researchers [16,37] to investigate how MLD within eddies affects the distribution of Chl-a and nitrate. Despite the seasonal variation in our data, the integrated Chl-a (I-Chl), SCM, and depth of SCM (DSCM) showed significant correlations with SLA (r = −0.58, −0.38, −0.27; Figure 10), all with $p < 0.01$. Similarly, integrated nitrate (I-nitrate) exhibited a strong negative correlation (r = −0.47, $p < 0.01$) with SLA (Figure 11a). These findings indicate that I-Chl, SCM, DSCM, and I-nitrate are more pronounced within CEs compared to the eddy edges, but weaker within ACEs. Furthermore, Figures 11, 12 and 13b demonstrate a significant correlation between I-Chl, SCM, DSCM, I-nitrate, and MLD (r = 0.45, −0.33, −0.21, 0.19; Figures 11b and 12), all with $p < 0.01$. Deeper MLD corresponds to decreased SCM and DSCM by factors of 0.005 $mg/m^4$ and 0.1765 m, respectively. Conversely, I-Chl and I-nitrate increase linearly by factors of 0.29 $mg/m^4$ and 0.75 µmol/L/m, respectively, with MLD deepening. Notably, most CEs (blue marks) exhibit larger SCM, DSCM, I-Chl, and I-nitrate values compared to ACEs (pink marks). Previous studies have demonstrated that CEs are associated with shallower euphotic depths than ACEs, providing more favorable light conditions for phytoplankton growth and leading to larger SCM and I-Chl values [14–16,38]. These findings highlight the critical role of nutrient supply in driving phytoplankton blooms.

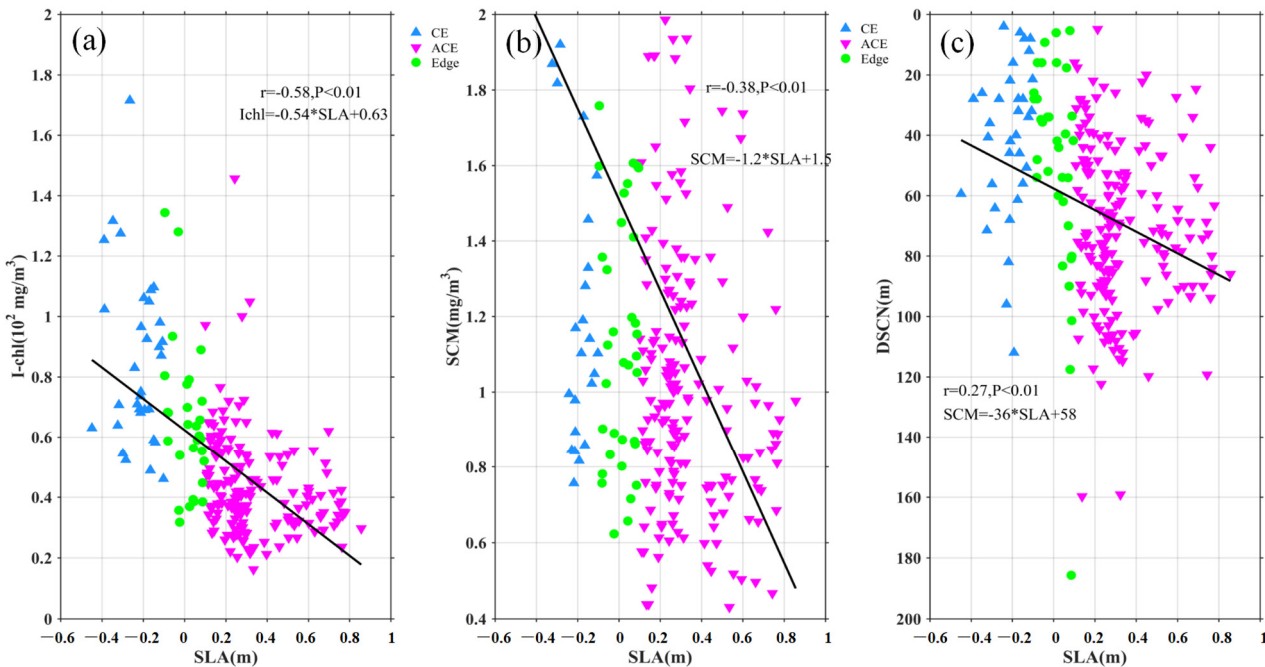

**Figure 10.** Statistical relationship between the depth-integrated Chl-a (**a**), SCM (**b**), DSCM (**c**), and SLA. The blue, pink, and green marks represent the observations in the CEs, ACEs, and edges, respectively. The black solid lines are the linear regressions.

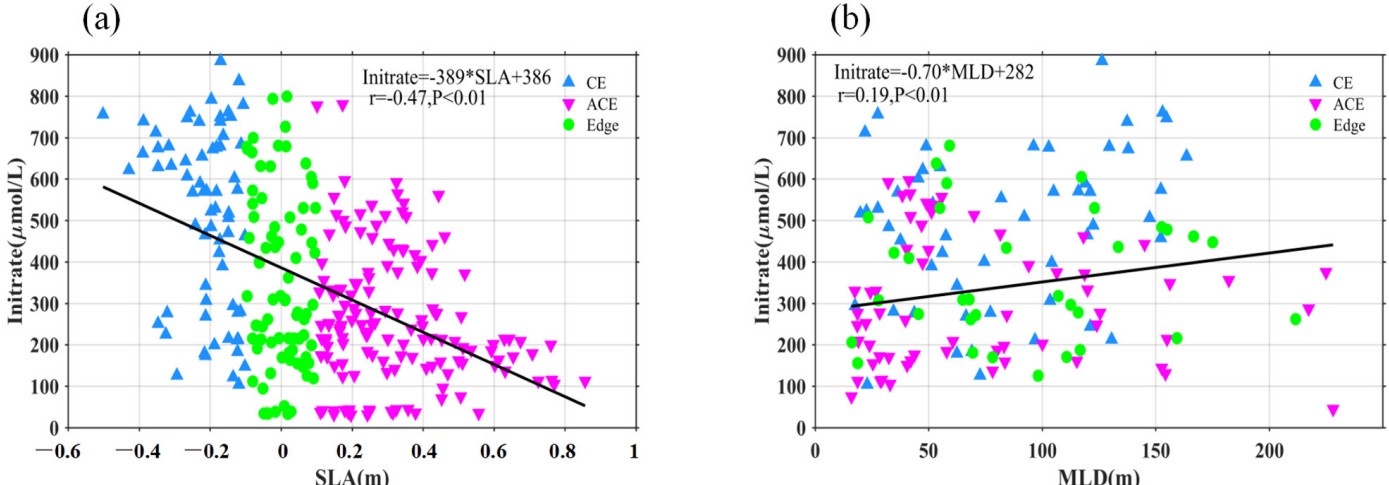

**Figure 11.** Statistical relationship between the depth-integrated nitrate and SLA (**a**) and MLD (**b**). The blue, pink, and green marks represent the observations in the CEs, ACEs, and edges, respectively. The black solid lines are the linear regressions.

The modulation of MLD within eddies plays a significant role in shaping the distribution of Chl-a. Generally, CEs (ACEs) enhance (dampen) nutrient supply through eddy-induced upwelling (downwelling), leading to high (low) Chl-a concentrations [5]. Previous studies have shown that eddy pumping is the dominant mechanism controlling the biogeochemical response in eddies, resulting in upward and downward displacement of the SCM layer [12]. However, other studies propose that ACEs are more productive than CEs in subtropical gyres due to winter deeper mixing (MLD), which enhances nutrient supply to the mixed layer and/or redistributes Chl-a through the stirring of the SCM [7,8,18]. In summer, the shallow MLD without strong vertical mixing does not promote the stirring of the SCM [16]. Figure 13 illustrates that changes in the MLD are influenced by turbulent

vertical mixing and adiabatic processes associated with vertical stretching and/or eddy pumping [18]. Eddy pumping leads to a deeper MLD in ACEs compared to CEs throughout the year, with the most significant differences observed in winter. Mesoscale eddies exhibit a shallow mixed layer with stable water (stronger $N^2$) throughout the year (Figure 13a,b). From summer to winter, as the mixed layer deepens, the shallow stratification dissipates, and the nitrate flux peaks by the end of winter. Subsequently, the water column undergoes restratification starting from spring. In CEs (ACEs), the summer stratification is stronger (weaker) than that in ACEs (CEs), resulting in a shallower (deeper) MLD, which facilitates a greater (smaller) injection of nutrients into the mixed layer during winter (Figure 13). However, these changes in MLD do not align with the occurrence of higher (lower) average Chl-a concentrations in CEs (ACEs). It is worth noting that negative (positive) Chl-a values in the core of CEs (ACEs) account for a significant proportion of the total CEs (ACEs) during winter (24% and 19%, respectively). This suggests that the differences in stratification during winter convective mixing, leading to variations in nutrient supply, may contribute to negative (positive) Chl-a concentrations in the core of CEs (ACEs). The opposite Chl-a phase can also be attributed to eddy stirring, which traps areas of high and low Chl-a concentrations, and/or eddy Ekman pumping [39].

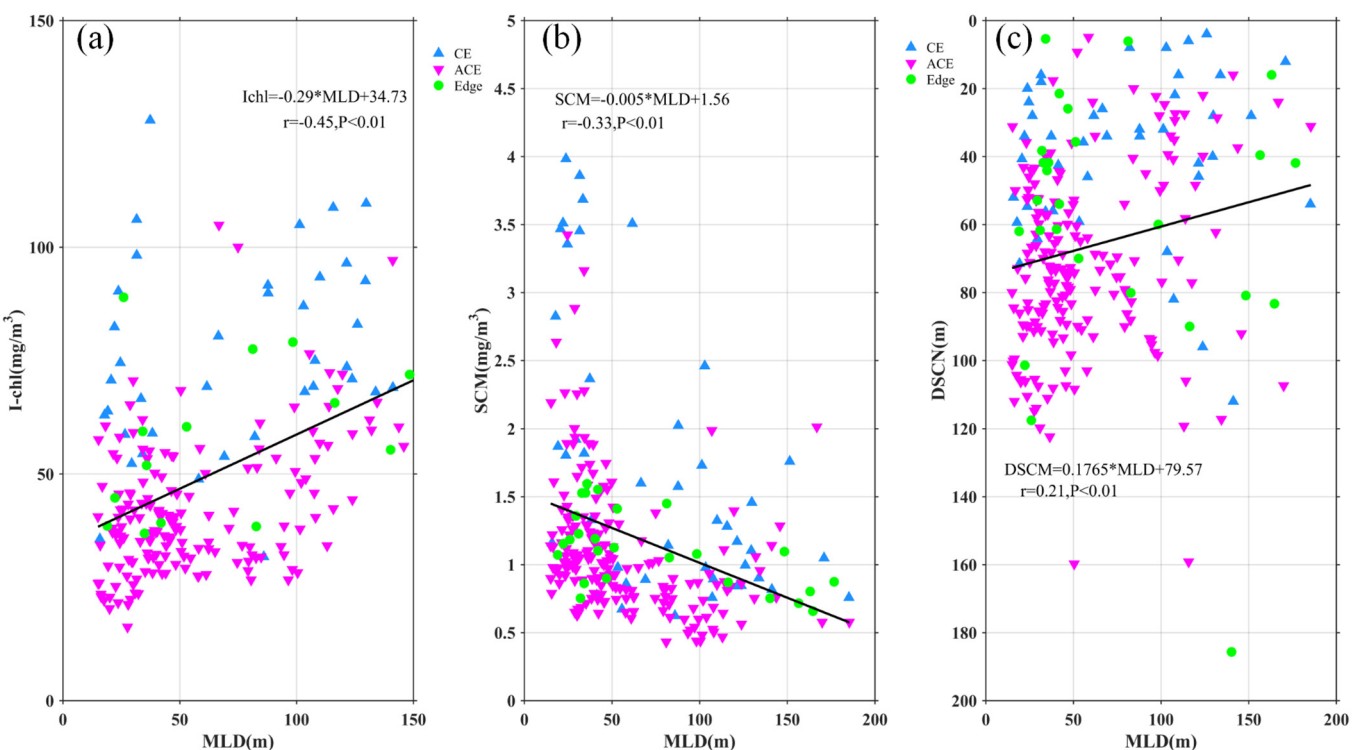

**Figure 12.** Statistical relationship between the depth-integrated Chl-a (**a**), SCM (**b**), DSCM (**c**), and MLD. The blue, pink, and green marks represent the observations in the CEs, ACEs, and edges, respectively. The black solid lines are the linear regressions.

### 4.2. Mesoscale Influence on Nutrient Budget

To assess the influence of mesoscale dynamics on biological production, the mixed layer nitrogen budget is analyzed using the OFES hydrography-biology products [9,30]. The nitrogen change in the OFES model (in mmol m$^{-2}$ d$^{-1}$) is divided into three components: mean advective flow, eddy flow, and mixing. During winter, the average nitrate increase is 2.67, with the three components contributing as follows: 0.54 (mean advective flow), −5.13 (eddy flow), and 7.26 (mixing) (Figure 14). In contrast, summer exhibits an average nitrate decrease of 4.64, with the three components contributing as follows: −0.34 (mean advective flow), −5.93 (eddy flow), and 1.64 (mixing) (Figure 15). These results indicate that eddy flows play a significant role in nutrient depletion, while the mean flow has a relatively

minor impact and contributes minimally to nutrient changes. The magnitude of these flows does not exhibit clear seasonal variation. On the other hand, the mixing component displays substantial variations in magnitude. Therefore, the seasonal variation of nitrate primarily depends on the influence of vertical mixing, highlighting the contribution of convective mixing processes to nutrient increase (winter) or decrease (summer) in the KE region.

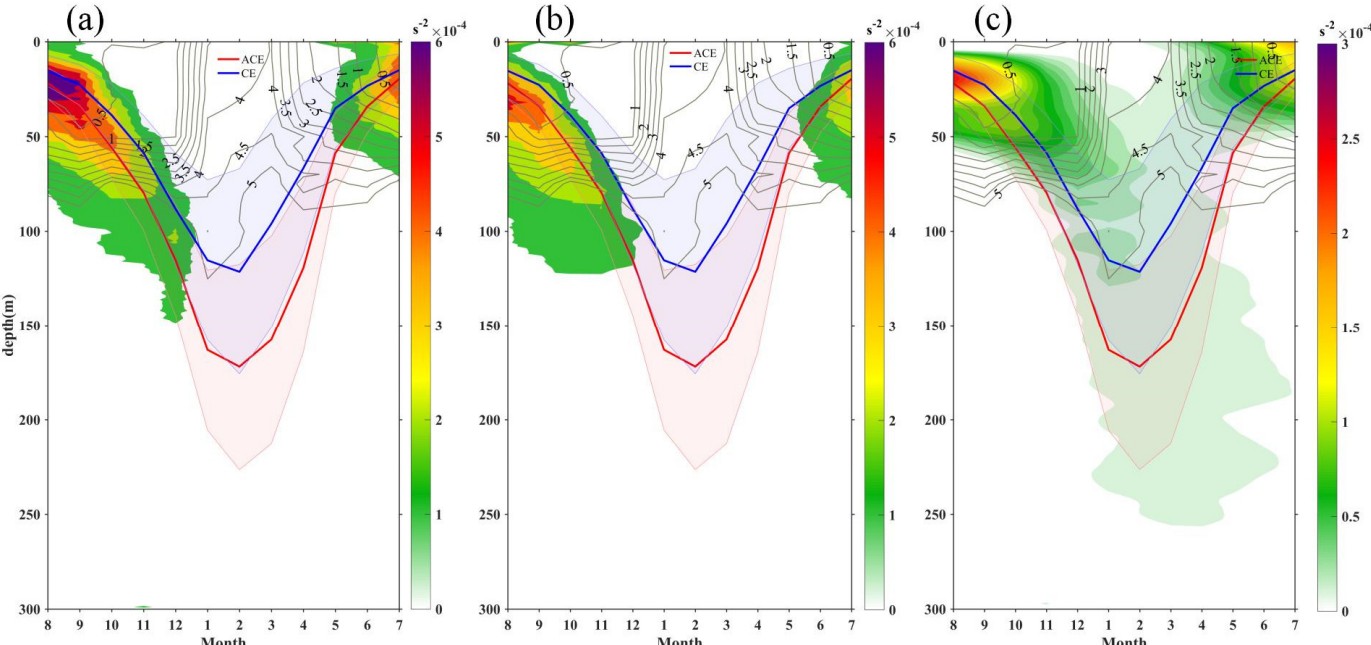

**Figure 13.** Buoyancy frequency (shading) and MLD (blue and red lines) from Argo floats within CEs (**a**) and ACEs (**b**) in the KE. (**c**) Buoyancy frequency difference between CEs and ACEs. The grey lines correspond to the nitrate mean seasonal contours.

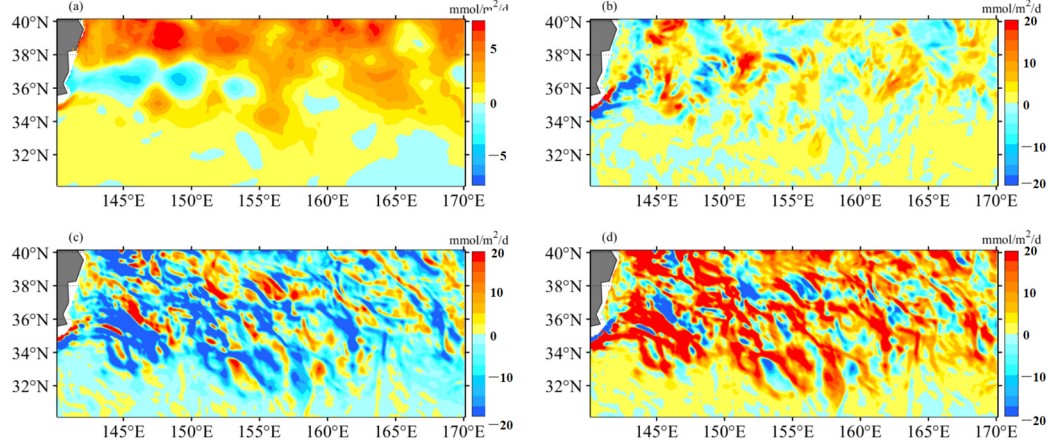

**Figure 14.** The nitrogen budget analysis using the OFES hydrography-biology products provides estimates of nitrate changes (in mmol m$^{-2}$ d$^{-1}$) within the mixed layer during the winter period from 1999 to 2009. Four components are considered: (**a**) the total change in nitrate, (**b**) the mean advective-induced component (horizontal + vertical), (**c**) the eddy-induced component (horizontal + vertical), and (**d**) the mixing component.

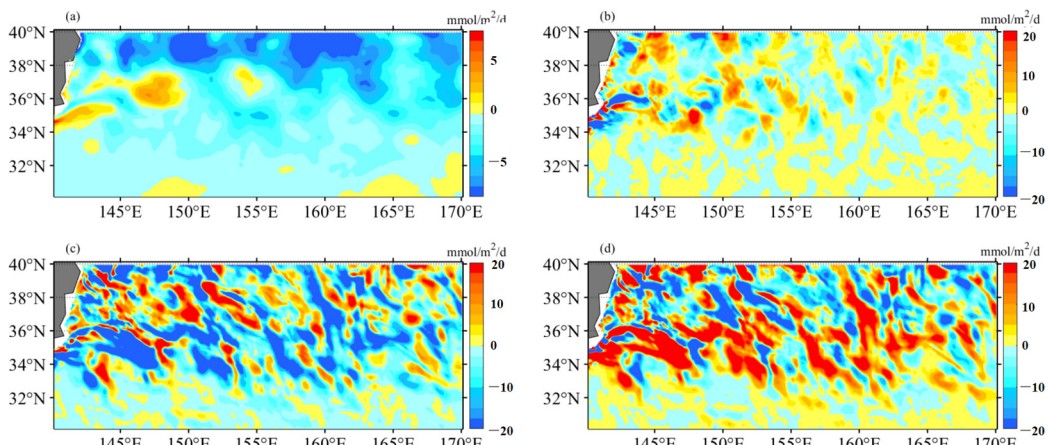

**Figure 15.** The nitrogen budget analysis using the OFES hydrography-biology products provides estimates of nitrate changes (in mmol $m^{-2}$ $d^{-1}$) within the mixed layer during the summer period from 1999 to 2009. Four components are considered: (**a**) the total change in nitrate, (**b**) the mean advective-induced component (horizontal + vertical), (**c**) the eddy-induced component (horizontal + vertical), and (**d**) the mixing component.

*4.3. Consideration of Submesoscale Processes*

It is important to consider the influence of submesoscale processes in the KE system, which are most pronounced during winter and spring [40]. Submesoscale features exhibit higher vertical velocities compared to eddies [41], making them important contributors to phytoplankton production by transporting nutrients and phytoplankton into the sunlit ocean [42–44]. This transport mechanism plays a significant role in promoting phytoplankton production. Although the analysis in this study may not fully capture the submesoscale features due to the spatial resolution of the data and potential obscuration in the EOF patterns, it is essential to acknowledge their potential influence for a comprehensive understanding of the biogeochemical dynamics in the KE system.

## 5. Conclusions

This study investigates the seasonality of eddy-induced Chl-a anomalies in the KE system. Using remote sensing and Argo floats data, we explore how eddies modify surface and subsurface Chl-a concentrations. The main conclusions are as follows:

1. CEs and ACEs exhibit opposite surface Chl-a anomalies, with CEs inducing positive anomalies and ACEs causing negative anomalies, particularly during winter. The monopole Chl-a patterns within the centers of the eddies correspond to positive or negative anomalies, depending on the sign of the principal component. These Chl-a anomalies account for approximately 26% and 18% of the total CEs and ACEs, respectively, across all seasons. These anomalies result from the uplifting or deepening of isopycnals and nitrate, stimulating or suppressing phytoplankton growth. Consequently, CEs and ACEs lead to variations in SCM depth-integrated Chl-a and nitrate, predominantly near the main axis of the KE.

2. The vertical distribution of Chl-a within eddies exhibits distinct patterns. Above the SCM layer, Chl-a concentrations are higher within CEs and lower within ACEs compared to the edge values, irrespective of winter variations. Conversely, below the SCM layer, Chl-a concentrations are lower within CEs and higher within ACEs than the edge values. Nutrient supply resulting from stratification differences under convective mixing and eddy stirring may contribute to these anomalies.

3. Additionally, another study examined the adjustment of MLD in eddies, revealing the influence of eddy-induced upwelling and downwelling in CEs and ACEs on nutrient supply and Chl-a concentrations. The differences between CEs and ACEs are more pronounced in winter due to deeper mixing, enhanced nutrient supply,

and the redistribution of Chl-a. The shallow mixed layer and stratification during summer affect nutrient injection and contribute to variations in Chl-a concentrations. Convective mixing processes also play a role in nutrient increase or decrease during winter and summer, respectively.

In conclusion, this study highlights the significance of mesoscale processes in driving biological productivity in the KE system. It provides valuable insights into the mechanisms underlying nutrient supply and Chl-a distributions. However, CEs generally have higher Chl-a in their centers due to submesoscale processes. It is important to note that the spatial resolution of observations limits a detailed exploration of submesoscale features using a statistical approach, such as higher vertical velocity in transporting nutrients and phytoplankton into the sunlit ocean and promoting significant phytoplankton production. Future research endeavors should prioritize improving spatial resolution and incorporating comprehensive observations to enhance our understanding of submesoscale features and their impacts.

**Author Contributions:** T.W. and S.Z. conceived and designed the study. T.W. helped with writing and provided thorough editing. T.W. and F.C. contributed to the writing and data interpretation. Data analyses were conducted mainly by T.W. and L.X. All authors have read and agreed to the published version of the manuscript.

**Funding:** This research was funded by the Scientific Research Start-Up Foundation of Lingnan Normal University (ZL22041).

**Data Availability Statement:** Not applicable.

**Acknowledgments:** The authors would like to thank the anonymous reviewers and the editor for their very constructive comments. The comments and suggestions provided by the reviewers and the editor have greatly enhanced the quality and presentation of this work. The authors would also like to extend their thanks to the Copernicus Marine Environmental Monitoring Center (CMEMS), the global Argo project, the National Oceanic and Atmospheric Administration (NOAA), and the Japan Agency for Marine-Earth Science and Technology (JAMSTEC) scientific teams, as well as D. Chelton and M. Schlax for their contributions in processing and providing the essential datasets used in this study.

**Conflicts of Interest:** The authors declare no conflict of interest.

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
