# Peer review of "The Seasonality of Eddy-Induced Chlorophyll-a Anomalies in the Kuroshio Extension System"

_remotesensing, doi:10.3390/rs15153865_

Round 1

Reviewer 1 Report

This manuscript investigated the seasonal variation of Chla anomaly in the Kuroshio extension. Few minor things.

Labels in most figures and tables are too small.

How did the authors define ‘edge’?

How did the authors define ‘MLD’?

Lines 138: Some words indicate Chl minus a

Figures 3: What are upper and lower panels?

Line 236: Cyclonic Eddy (CE)

Line 425: Mesoscale eddies

Line 490-492: repeating sentences

This manuscript lacks evidence of submesoscale processes impacting Chla variation, yet it highlights their importance in the conclusion, based on assumptions.

Reviewer 3 Report

Please read the attached file.

Round 2

Reviewer 1 Report

All my concerns are addressed. I have no objection to the publication. 

Reviewer 2 Report

The authors misunderstood my suggestion about the conclusion. The conclusion should be restructured into one paragraph with four sentences.

Reviewer 3 Report

The paper has been revised as the suggestions.